# Burden and predictors of anxiety disorder among HIV patients on ART in Nairobi Kenya

**Kevin Kibera Gacau**[1]*, **George Mugendi**[2], **George Kiragu**[3], **Musa Otieno Ngayo**[4], **Gloria Omosa**[5]

1 Department of Paediatrics and Child Health, University of Nairobi, Nairobi, Kenya, 2 Department of Pharmacy, University of Nairobi, Nairobi, Kenya, 3 Department of Pharmacy, Directorate of Clinical Services, Kiambu County, Nairobi, Kenya, 4 Centre of Microbiology Research, Kenya Medical Research Institute, Nairobi, Kenya, 5 Department of Medical Microbiology and Immunology, University of Nairobi, Nairobi, Kenya

* kkibera.g@gmail.com

**Data Availability Statement:** The data has been uploaded as supplementary information.

**Funding:** KKG; Research work was funded by Health-Professional Education Partnership Initiative

## Abstract

Anxiety disorders are prevalent among people living with Human Immunodeficiency Virus (HIV) globally, but data on the prevalence and factors associated with this mental illness are limited among HIV patients on antiretroviral therapy (ART) in Kenya. This study determined the prevalence and correlates of anxiety disorder among HIV patients receiving care and treatment at the Comprehensive Care Clinic (CCC) in Mbagathi Hospital in Nairobi, Kenya. This was a cross-sectional study where 205 eligible and consenting participants were systematically enrolled. The Beck Anxiety Inventory (BAI) Questionnaire was used to assess anxiety levels, with a score of 8 or more indicating moderate to severe anxiety. Structured questionnaires were used to collect information on factors associated with anxiety disorder. Descriptive statistics and logistic regression models were used to analyze data. Of the enrolled 205 patients, 122 (59.5%) were female. The median age and household size were 49 years (Interquartile range IQR 39–54) and 3 people (IQR 2–3) respectively. A large proportion of the participants, 169 (82.4%) were on the first-line ART regimen and the median duration on ART was 13 years (IQR 7–18). Nearly a third of the participants 60 (29.3%) had anxiety disorder. Male participants (adjusted odds ratio—aOR 0.33; 95% confidence interval—CI 0.13–0.78) were less likely to have anxiety disorder. Self-employment (aOR 5.57; 95% CI 2.17–20.06), larger households (aOR 1.48; 95%CI 1.11–2.03) and no history of tuberculosis infection (aOR 2.9; 95% CI; 1.29–6.96) were factors associated with having anxiety disorder. Among PLHIV receiving ART in Nairobi County, Kenya, there is a considerable burden of anxiety disorder which was associated with gender, occupation, household population size and history of tuberculosis infection.

## Introduction

Anxiety is a relatively common mental disorder, more so among people living with HIV (PLHIV) [1]. PLHIV are twice as likely as the general population to be diagnosed with an anxiety disorder (mental disorders) [2]. The relationship between HIV and mental health seems bidirectional. It has been suggested that a HIV diagnosis may increase the likelihood of mental

(HEPI)-Kenya a National Institute of health (NIH) Funded Project-R25TW011212. The funders had no role in study design, data collection and analysis, decision to publish, or preparation of the manuscript.

**Competing interests:** The authors have declared that no competing interests exist.

disorders among infected individuals and that the progression of HIV may be hastened by mental disorders like anxiety and depression [1, 3].

Globally, the average lifetime prevalence of anxiety disorder is reported as 29% [1]. The prevalence of anxiety disorder among PLHIV ranges between 19% to 49% in developing and developed nations such as Mexico, Brazil, United Kingdom and North America [4–6]. Regionally, a few studies have reported burdens of between 14–34% in Guinea and Ethiopia. However, a dearth exists of similar data in sub–Saharan Africa as a whole, Kenya included. Some of the important correlates for anxiety disorder identified among PLHIV includes; age, socioeconomic status, household population size, virologic and immunological comorbidities [7–12]. Studies providing data on correlates among PLHIV in sub–Saharan Africa are few.

Although mental health is a universal human right according to World Health Organization (WHO), the mental health of PLHIV is often neglected and still need more attention and interest as a public health concern [13]. In low- and middle-income countries, mental health among PLHIV is only beginning to receive the deserving attention [14]. This implies that the burden of anxiety and the associated factors are neither well known nor understood [15].

To help bridge this research gap in Kenya, we conducted a cross sectional study to determine the prevalence among PLHIV receiving treatment at Mbagathi Hospital in Nairobi County. Furthermore, as a secondary aim, we evaluated predictors correlated with anxiety among this population.

## Materials and methods

### Study design and setting

This cross-sectional study was conducted between 1st February, 2023 and 31st May, 2023 among PLHIV receiving Antiretroviral therapy (ART) at Mbagathi Hospital, Nairobi County. This facility is a 200-bed hospital that offers essential curative and preventive healthcare services. The facility is located in Nairobi County less than a 20-minute drive from the Nairobi Central Business District and less than a 10-minute drive from the National referral hospital, Kenyatta National hospital. The hospital with a workforce of approximately 500 staff members serves as a primary health facility for the heterogeneous population around it and is also a referral hospital for levels 2 and 3 facilities within the county. The facility is a teaching hospital for Kenya Medical Training College and other Kenyan Universities. According to the March 2022 facility records, the Mbagathi hospital Comprehensive Care Clinic (CCC) has enrolled approximately 4700 patients (children and adults), reviews 80–100 patients daily, and has an average of 1300 patients monthly.

Eligible participants were those that tested HIV positive according to the Kenyan guideline and were enrolled into HIV care and treatment program, aged $\geq$ 18 years, attending the Mbagathi Hospitals' CCC, and who voluntarily provided written informed consent. Exclusion criteria included those eligible but declined to provide consent. The sample size was calculated to be 183 using the formula described by Lachenbruch et al. [16], setting α at 0.05 and considering an anxiety prevalence of 13.8% among PLHIV [17]. To account for potential incomplete data, we enrolled an additional 12% of participants. Systematic random sampling was employed, with invitations extended to every third patient attending the HIV clinic during the study period. The selection interval was determined by dividing the total number of patients attending the clinic during the study duration by the desired sample size.

### Ethical statement

This study was done according to the principles of the Declaration of Helsinki and was approved by the University of Nairobi / Kenyatta National Hospital Ethical Review Committee

(KNH-ERC/A/413) and was granted the National Commission for Science, Technology and Innovation (NACOSTI) License (No: NACOSTI/P/22/21476). Before recruitment in this study, according to the inclusion criteria, all participants filled in a written informed consent for study participation.

## Data collection

**Data collection instruments.** The data collection process was planned and organized as guided by the CCC's medical in-charge. The best time and day for questionnaire administration were identified to suite each of the enrolled HIV patients. Two of the hospital's clinical psychologists were trained to administer the questionnaires to the respondents. These research assistants contacted the consenting HIV patients explaining the purpose and rational of the study. These research assistants collected all the study related data using face- to-face interview. Each respondent took approximately 30 minutes to complete the questionnaire.

Anxiety levels were assessed using the BAI, a validated 21-item self-report inventory. The BAI measures the severity of anxiety in adults and adolescents, with scores ranging from 0 to 63. BAI scores were interpreted as follows: 0 to 7 = no/minimal anxiety, 8 to 15 = mild anxiety, 16 to 25 = moderate anxiety, and 26 to 63 = severe anxiety [18]. Sociodemographic and socioeconomic data, including age, gender, marital status, employment status, household size, and monthly income, were collected using structured interviewer-administered questionnaires. Clinical data, such as the type of ART regimen, duration on ART, current Clusters of differentiation 4 (CD4) count, viral load, and the presence of other comorbidities, were also collected.

## Statistical analyses

Descriptive statistics summarized the sociodemographic, socioeconomic, and clinical data. The median and interquartile range (IQR), was used for numeric variables (age and household size) to minimize the effects of outliers and skewness in the data. The frequency counts and percentages were used to express all categorical variables. The proportion of participants with a BAI score of 8 or more, indicating moderate to severe anxiety, was calculated. Bivariate logistic regression analyses assessed the association of each variable (demographic and clinical) with the presence of symptoms related with moderate to severe anxiety. A multivariate logistic regression model explored the independent associations between identified factors and anxiety levels. The variables included in the multivariate logistic regression model were purposively selected as described by Lachenbruch et al. [16]. The variables that were first significant with a Wald test were selected. Previous research established that a cut-off p-value of 0.05 for variable selection is usually too stringent and leads to the exclusion of otherwise important variables [16, 19]. As a result, several studies have suggested a more 'lenient' range for variable selection (0.15–0.4) [16, 19]. Following various simulations during the analysis, the cut-off p-value of 0.2 was settled upon as it best fit the data. This cut-off value was only used during selection of variables to be included in the multivariate model. The assessments of level of significance during the univariate and multivariate analysis, were however set at $P \leq 0.05$. Data analysis was performed using R software version 4.1.2.

## Results

A total of 205 participants with a median age of 49 years (IQR 39–54), were enrolled in the study). The majority of the study participants were female (122, 59.5%) while 113(55.1%) were on contractual or pensionable employment. The median household size was 3 people (IQR 2–3) and a large proportion of the participants 169 (82.4%) were on the first-line ART regimen, and their median duration on ART was 13 years (IQR 7–18). Viral suppression (less than

50 copies/mL) was observed in 168 (81.9%) of the participants, and 191 (93.2%) had a CD4 count of 500 cells/mm3 or more. Despite this, 63 (30.7%) of the participants had a history of tuberculosis infection, and 35 (17.1%) had other comorbidities (Table 1).

Using the BAI 21 screening tool, 60 (29.3%) patients with varying severities of anxiety were identified; 43(21%) had mild anxiety, 14(6.8%) had moderate anxiety, and 3(1.5%) had severe anxiety (Fig 1). The proportion of participants with a BAI score of 8 or more indicated mild to severe anxiety. Using this categorization; there were 60 (29.3%) of the participants who were considered anxious (including mild, moderate and severe anxiety levels) compared to 145 (70.7%) of the participants without the presence of anxiety disorder. The median BAI score was 4 (IQR 1–9) and the individual symptoms score are listed in Table 2.

Univariate analysis of selected baseline characteristics and presence of anxiety revealed that a significant connection between marital status and anxiety, whereby both genders who were never married or divorced had higher levels of anxiety (p = 0.011). Similarly, analysis of employment status showed that the self-employed groups had higher anxiety levels (p < 0.001). Also, high viral loads were associated with increased anxiety levels (p = 0.026; Table 3).

**Table 1. Baseline characteristics of the study participants (n = 205 patients).**

| Variables | | n (%) |
|---|---|---|
| *Age in years; median (IQR)* | | 49 (39–54) |
| *Sex* | Female | 122 (59.5) |
| | Male | 83 (40.5) |
| *Highest level of Education obtained* | No formal education | 13 (6.3) |
| | Primary school | 52 (25.4) |
| | Secondary school | 29 (14.2) |
| | College/University | 111 (54.2) |
| *Marital status* | Never Married | 32 (15.6) |
| | Married | 109 (53.2) |
| | Separated/divorced/widowed | 64 (31.2) |
| *Employment status* | Unemployed | 40 (19.5) |
| | Contractual/pensionable Employment | 113 (55.1) |
| | Self-Employment | 52 (25.4) |
| *Spouse Occupation* | Unemployed | 112 (54.6) |
| | Salaried Job/casual laborer | 54 (26.3) |
| | Self-Employed | 39 (19.1) |
| *Household Size; median (IQR)* | | 3 (2–3) |
| *ART Regimen* | First Line | 169 (82.4) |
| | Second & Third Line | 36 (17.6) |
| *Number of years on ART; median (IQR)* | | 13 (7–18) |
| *Latest Viral Load* | < 50 | 168 (81.9) |
| | ≥ 50 | 37 (18.5) |
| *Latest CD4 Count* | ≥ 500 | 191 (93.2) |
| | < 500 | 14 (6.8) |
| *TB History* | Yes | 63 (30.7) |
| | No | 142 (69.3) |
| *Comorbidity* | Yes | 35 (17.1) |
| | No | 170 (82.9) |

n–number of patients; IQR—interquartile range

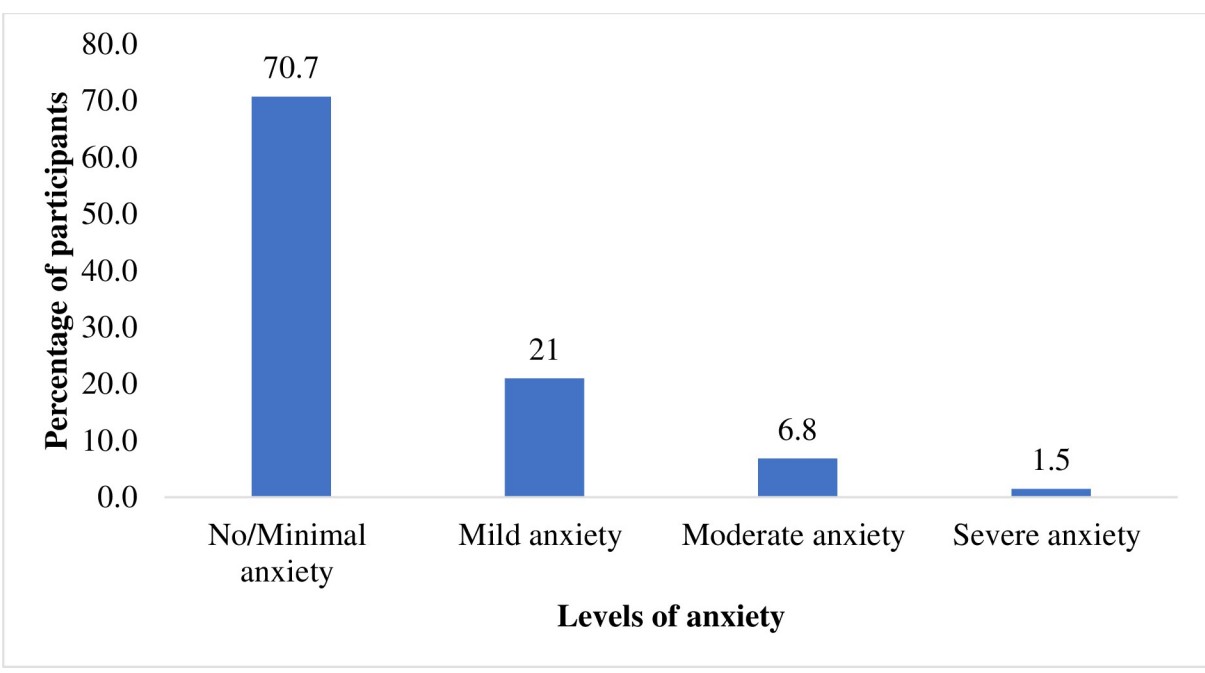

**Fig 1. Distribution of anxiety level based on four group categorizations of Beck Anxiety Inventory.**

In the multivariable model, male participants were less likely to have anxiety compared to the females (aOR 0.33; 95%CI 0.13–0.78, p = 0.014). Self-employed participants were approximately 6 times more likely to have anxiety as opposed to those who were unemployed (aOR 5.57; 95% CI 2.17–20.06, p = 0.006). Individuals from households larger than two members were 48% more likely to have anxiety compared to those from families with household comprising of one individual (aOR 1.48; 95%CI 1.11–2.03, p <0.001). Lastly, patient who had no history of tuberculosis infection were 2.90 times more likely to have anxiety compared to the participants who had history of tuberculosis infection (aOR 2.9; 95% CI; 1.29–6.96, p = 0.013) (Table 4).

## Discussion

This study found that 29.3% of PLHIV receiving ART treatment had anxiety disorder categorized using BAI. The severity of anxiety varied among participants, with a majority (21%) having mild anxiety yet moderate (6.8%) and severe (1.5%) anxiety were also reported among this population. Factors such as gender, employment status, household size, and history of tuberculosis infection were significantly associated with the presence of anxiety.

The PLHIV face a series of mental health challenges including anxiety [1, 2]. The prevalence of anxiety among PLHIV in sub-Saharan Africa exhibits considerable variation across countries. For instance, studies have reported prevalence ranging from as low as 9% in South Africa [20] to as high as 34.4% in Ethiopia [21]. Specifically, a study in coastal Kenya identified a prevalence of 11.7% [10], while in Nigeria and Guinea, the reported prevalence was 32.6% [22] and 13.8% [17], respectively. Furthermore, in a broader context, certain low middle income countries regions have even reported anxiety prevalence exceeding 30% among PLHIV [23]. The observed variation in prevalence of anxiety among PLHIV between the current study and others may stem from differences in study designs and sample sizes [10]. When making cross-cultural comparisons, it's notable that prevalence can be highly diverse. While this heterogeneity

**Table 2. Itemized scores for the 21 items on the Beck's Anxiety Inventory (n = 205).**

| | How much you have been bothered by that symptom during the past month, including today? | Not at all (n, %) | Mildly, but it didn't bother me much (n, %) | Moderately-it wasn't pleasant at times (n, %) | Severely-it bothered me a lot (n, %) |
|---|---|---|---|---|---|
| 1 | Numbness or tingling | 113 (55.1) | 66 (32.2) | 20 (9.8) | 6 (2.9) |
| 2 | Feeling hot | 131 (63.9) | 61 (29.8) | 12 (5.9) | 1 (0.5) |
| 3 | Wobbliness in legs | 142 (69.3) | 50 (24.4) | 11 (5.4) | 2 (0.9) |
| 4 | Unable to relax | 140 (68.3) | 50 (24.4) | 12 (5.9) | 3 (1.5) |
| 5 | Fear of worst happening | 157 (76.6) | 44 (21.5) | 4 (1.9) | 0 (0.0) |
| 6 | Dizzy or lightheaded | 127 (61.9) | 66 (32.2) | 12 (5.9) | 0 (0.0) |
| 7 | Heart pounding/racing | 135 (65.9) | 56 (27.3) | 12 (5.9) | 2 (0.9) |
| 8 | Unsteady | 159 (77.6) | 43 (20.9) | 3 (1.5) | 0 (0.0) |
| 9 | Terrified or afraid | 176 (85.9) | 27 (13.2) | 2 (0.9) | 0 (0.0) |
| 10 | Nervous | 121 (59) | 73 (35.6) | 10 (4.9) | 1 (0.5) |
| 11 | Feeling of choking | 199 (97.1) | 4 (1.9) | 2 (0.9) | 0 (0.0) |
| 12 | Hands trembling | 171 (83.4) | 31 (15.1) | 3 (1.5) | 0 (0.0) |
| 13 | Shaky/Unsteady | 166 (80.9) | 36 (17.6) | 3 (1.5) | 0 (0.0) |
| 14 | Fear of losing control | 165 (80.5) | 38 (18.5) | 2 (0.9) | 0 (0.0) |
| 15 | Difficulty in breathing | 187 (91.2) | 16 (7.8) | 2 (0.9) | 0 (0.0) |
| 16 | Fear of dying | 146 (71.2) | 51 (24.9) | 7 (3.4) | 1 (0.5) |
| 17 | Scared | 166 (80.9) | 37 (18.1) | 2 (0.9) | 0 (0.0) |
| 18 | Indigestion | 159 (77.6) | 37 (18.1) | 8 (3.9) | 1 (0.5) |
| 19 | Faint/lightheaded | 167 (81.5) | 37 (18.1) | 1 (0.5) | 0 (0.0) |
| 20 | Face flushed | 187 (91.2) | 18 (8.8) | 0 (0.0) | 0 (0.0) |
| 21 | Hot/cold sweats | 153 (74.6) | 49 (23.9) | 3 (1.5) | 0 (0.0) |

might initially seem attributable to cultural influences, it is more plausible that methodological differences play a significant role [10]. However, genuine disparities between the populations studied cannot be ruled out. Such disparities might arise from genetical differences across races and ethnic groups [24, 25], culturally influenced psychosocial factors (such as distinct societal roles for women), or traumatic stressors that impact entire nations or ethnic groups [10].

Male patients were less likely to have anxiety disorder consistent with findings from other settings among PLHIV [26]. Similar to the current study, Tesfaw et al., [27] reported male sex

**Table 3. Comparative analysis of selected baseline sociodemographic traits and anxiety (n = 205 patients).**

| Variable | Presence of Anxiety (n = 205) | | P–Value* |
|---|---|---|---|
| | Yes (n = 60) (n, %) | No (n = 145) (n, %) | |
| **\|Sex** | | | 0.052 |
| Female | 42 (70) | 80 (55.2) | |
| Male | 18 (30) | 65 (44.8) | |
| **Highest level of education obtained** | | | 0.084 |
| No formal education | 8 (13.3) | 5 (3.5) | |
| Primary school | 15 (25) | 37 (25.5) | |
| Secondary school | 6 (10) | 23 (15.9) | |
| College/University | 31 (51.7) | 80 (55.1) | |
| **Marital status** | | | **0.011** |
| Never Married | 8 (13.3) | 24 (16.6) | |
| Married | 24 (40) | 85 (58.6) | |
| Separated/divorced/widowed | 28 (46.7) | 36 (24.8) | |
| **Employment status** | | | **<0.001** |
| Unemployed | 7 (11.7) | 33 (22.8) | |
| Contractual/pensionable | 26 (43.3) | 87 (60) | |
| Self-employment | 27 (45) | 25 (17.2) | |
| **Spouse Occupation** | | | 0.054 |
| Unemployed | 36 (60) | 76 (52.4) | |
| Salaried/casual laborer | 9 (15) | 45 (31) | |
| Self-employed | 15 (25) | 24 (16.6) | |
| **ART Regimen** | | | 0.537 |
| First Line | 51 (85) | 118 (81.4) | |
| Second & Third Line | 9 (15) | 27 (18.6) | |
| **Latest Viral Load** | | | **0.026** |
| < 50 | 55 (91.7) | 113 (77.9) | |
| ≥ 50 | 5 (8.3) | 32 (22.1) | |
| **Latest CD4 Count** | | | 0.219 |
| ≥ 500 | 58 (96.7) | 133 (91.7) | |
| < 500 | 2 (3.3) | 12 (8.3) | |
| **TB History** | | | 0.144 |
| Yes | 14 (23.3) | 49 (33.8) | |
| No | 46 (76.7) | 96 (66.2) | |
| **Comorbidity** | | | 0.13 |
| Yes | 14 (23.3) | 21 (14.5) | |
| No | 46 (76.7) | 124 (85.5) | |

*Chi-Square Test

had decreased levels of anxiety among PLHIV. It is postulated that the diverse approaches between the sexes to seek social support and disclosure during HIV affects how anxious one becomes [28]. Females living with HIV from sub-Saharan Africa could be at a higher risk of anxiety disorder because of additional experiences of traumatic events such as sexual abuse and intimate partner violence, some of which may have had a role in their acquisition of HIV infection [29]. Additionally, they are more likely to be stigmatized and blamed for HIV transmission within families in patriarchal families, societies and communities [30].

**Table 4. Bivariable and multivariable logistic regression analysis for the covariates of anxiety (n = 205 patients).**

| Variables | Bivariate | | Multivariate | |
|---|---|---|---|---|
| | cOR (95%CI) | P—value | aOR (95%CI) | P—value |
| **Age (Years)** | 1.04 (1.01–1.07) | **0.006** | 1.01 (0.97–1.05) | 0.716 |
| **\|Sex** | | | | |
| Female | Referent | | Referent | |
| Male | 0.53 (0.27–0.99) | **0.051** | 0.33 (0.13–0.78) | **0.014** |
| **Highest level of education obtained** | | | | |
| No formal education | Referent | | Referent | |
| Primary school | 0.25 (0.07–0.88) | **0.034** | 0.54 (0.11–2.54) | 0.436 |
| Secondary school | 0.16 (0.04–0.66) | **0.013** | 0.32 (0.05–1.80) | 0.201 |
| College/University | 0.24 (0.07–0.78) | **0.019** | 0.41 (0.09–1.76) | 0.232 |
| **Marital status** | | | | |
| Never Married | Referent | | Referent | |
| Married | 0.85 (0.35–2.23) | 0.724 | 0.31 (0.05–1.63) | 0.176 |
| Separated/divorced/widowed | 2.33 (0.94–6.26) | 0.077 | 1.06 (0.29–4.05) | 0.926 |
| **Employment status** | | | | |
| Unemployed | Referent | | Referent | |
| Contractual/pensionable Employment | 1.41 (0.58–3.80) | 0.468 | 2.20 (0.77–6.99) | 0.156 |
| Self-employment | 5.09 (1.99–14.44) | **0.001** | 5.57 (2.17–20.06) | **0.006** |
| **Spouse Occupation** | | | | |
| Unemployed | Referent | | Referent | |
| Salaried/Casual laborer | 0.42 (0.18–0.92) | **0.039** | 0.87 (0.21–3.80) | 0.847 |
| Self-employed | 1.32 (0.62–2.80) | 0.473 | 3.62 (0.92–16.50) | 0.077 |
| **Household Size** | 1.44 (1.17–1.82) | **0.001** | 1.48 (1.11–2.03) | **<0.001** |
| **Latest Viral Load** | | | | |
| < 50 | Referent | | Referent | |
| ≥ 50 | 0.32 (0.11–0.80) | **0.025** | 0.4 (0.10–1.28) | 0.151 |
| **TB History** | | | | |
| Yes | Referent | | Referent | |
| No | 1.68 (0.86–3.43) | 0.142 | 2.9 (1.29–6.96) | **0.013** |
| **Comorbidity** | | | | |
| Yes | Referent | | Referent | |
| No | 0.55 (0.26–1.20) | 0.129 | 0.46 (0.18–124) | 0.122 |

In this study, self-employment was associated with having anxiety disorder. The study postulate that the uncertainty of the environment and hard economic times is a trigger for anxiety more so among PLHIV. Consistent with this study Maier *et al*., [9], associates higher monthly household income with lower anxiety disorder among PLHIV. The importance of socioeconomic status both at the household and societal level has previously been associated with anxiety episodes [10, 31].

The current study's findings resonate with the broader regional context, though direct comparisons with other sub-Saharan Africa countries are limited due to the scarcity of data, especially in Kenya. The observed association between larger household sizes is aligned to economic empowerment and increased anxiety risk aligns with previous research further strengthening the role of socio-economic factors and mental health outcomes among PLHIV [9, 10, 27].

In the current study, participants from household with large population sizes (median IQR) were more likely to have anxiety disorder. Studies evaluating the role of family size and anxiety

disorder are particularly rare among PLHIV. However, previous reports among the general population are conflicting, some show association between large family size and increased risk of anxiety disorder [32], while others showing large family/household size being protective [33]. Household population size is an indicator of both socioeconomic status as well as availability of social support [34]. Anxiety disorders especially in large family sizes from poor and rural setting are thought to relate to limited resources rather than the family size [32]. In China, anxiety disorder is shown to relate to poor family functioning and quality of life, as well as lack of social support which were more prevalent in smaller families [35].

This study showed that previous tuberculosis (TB) co-infection is associated with lower prevalence of anxiety disorder. In contrast, in Ethiopia, Deribew *et al.*, [36] showed that TB and HIV co-infected PLHIV had significantly greater risk of anxiety disorder than the non-co-infected patients. In South Africa, higher prevalence of anxiety disorder was significantly associated with tuberculosis co-infection [37]. HIV and TB coinfected individuals are at greater risk for relapse following treatment as well as increased chances of drug resistance to anti-TB drugs [38]. Tuberculosis co-infection for an HIV-infected patients means more pills, and more strict observation because tuberculosis not only accelerates the deterioration of the immune status of patients with HIV, it is one of the leading causes of mortality in PLHIV [39]. Further, HIV medications including stavudine, zidovudine and tuberculosis medications including isoniazid have been known to cause anxiety symptoms by themselves [40]. In Kenya, patients diagnosed with TB and HIV co-infection are started on anti- tuberculosis treatment immediately and while ART is initiated as soon as anti- tuberculosis medications are tolerated [41, 42]. For patients with TB/HIV co-infection who are already on ART are started on anti- tuberculosis treatment immediately and continue ART, making any required adjustments to the ART regimen based on known drug-drug interactions and monitoring toxicity [41, 42]. These treatment guidelines are generally subjected on directly observed therapy. During directly observed therapy, patients are exposed to a significant emotional support and increased clinical monitoring which have been shown to help patients complete treatment and limit mental health issues. We postulate that PLHIV in our study who had previous tuberculosis coinfection had learnt and adopted better coping strategies during the treatment of both HIV and tuberculosis compared to those who had HIV only. This is in agreement with previous studies that reported the benefits of surviving management of co-infection among PLHIV and how this prepares them mentally for any comorbidity due to HIV infection [37–40].

The current study was not without limitations. First, due to the cross-sectional nature, the study was unable to infer causality. Additionally, while the study captured a range of socio-demographic and clinical variables, the study did not evaluate other psychosocial determinants such as substance use, or access to mental health services. However, a robust methodology and the comprehensive assessment of various potential correlates were significant strengths of this study. Secondly, depression and anxiety are often comorbid among PLHIV, and their symptoms can overlap, impacting the accuracy of anxiety assessments. While this study focused on anxiety, it is acknowledged that assessing for depression could provide a more comprehensive understanding of mental health in this cohort. In summary, this study underscores the significant burden of anxiety among PLHIV receiving ART treatment in Nairobi. The identified correlates in this study emphasized the multifaceted nature of anxiety and the need for a holistic approach to care that integrates mental health services within HIV care settings.

## Supporting information

**S1 Checklist.**
(DOC)

**S1 Data.**
(XLSX)

## Acknowledgments

The authors wish to acknowledge all the study patients and staff of Mbagathi Hospital, Nairobi County. Further, we acknowledge the training support from the University of Nairobi's Building Capacity for Writing Scientific Manuscripts (UANDISHI) Program at the Faculty of Health Sciences.

## Author Contributions

**Conceptualization:** Kevin Kibera Gacau, George Mugendi.

**Data curation:** George Kiragu, Musa Otieno Ngayo.

**Formal analysis:** George Kiragu, Musa Otieno Ngayo.

**Funding acquisition:** Gloria Omosa.

**Methodology:** Kevin Kibera Gacau.

**Project administration:** Kevin Kibera Gacau, Gloria Omosa.

**Resources:** Gloria Omosa.

**Supervision:** George Mugendi.

**Validation:** Musa Otieno Ngayo.

**Writing – original draft:** Kevin Kibera Gacau.

**Writing – review & editing:** Kevin Kibera Gacau, George Mugendi, Musa Otieno Ngayo, Gloria Omosa.

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
