## [Decision Letter · Decision Letter 0]

7 Feb 2024

PMEN-D-23-00061

Anxiety disorder and predictors among HIV patients in Nairobi Kenya

PLOS Mental Health

Dear Dr. Gacau,

Thank you for submitting your manuscript to PLOS Mental Health. After careful consideration, we feel that it has merit but does not fully meet PLOS Mental Health’s publication criteria as it currently stands. Therefore, we invite you to submit a revised version of the manuscript that addresses the points raised during the review process.

We look forward to receiving your revised manuscript.

Kind regards,

Matteo Monzio Compagnoni, Ph.D.

Academic Editor

PLOS Mental Health

Journal Requirements:

1. Please amend your detailed online Financial Disclosure statement. This is published with the article. It must therefore be completed in full sentences and contain the exact wording you wish to be published.

a) State the initials, alongside each funding source, of each author to receive each grant. For example: "This work was supported by the National Institutes of Health (####### to AM; ###### to CJ) and the National Science Foundation (###### to AM)."

2. Please update your online Competing Interests statement. If you have no competing interests to declare, please state: “The authors have declared that no competing interests exist.”

3. In the online submission form, you indicated that "Data will be provided by the corresponding author upon request".

a) In a public repository, 

b) Within the manuscript itself, or 

c) Uploaded as supplementary information.

4. Please provide separate figure files in .tif or .eps format only and remove any figures embedded in your manuscript file. Please also ensure that all files are under our size limit of 10MB. You may leave the figure captions or legends in the manuscript.

For more information about how to convert your figure files please see our guidelines: https://journals.plos.org/mentalhealth/s/figures

5. We do not publish any copyright or trademark symbols that usually accompany proprietary names, eg (R), (C), or TM (e.g. next to drug or reagent names). Please remove all instances of trademark/copyright symbols throughout the text, including ® on pages 2 and 5.

6. We have noticed that you have uploaded Supporting Information files, but you have not included a list of legends. Please add a full list of legends for your Supporting Information files after the references list.

Additional Editor Comments (if provided):

I’ve read the work titled “Anxiety disorder and predictors among HIV patients in Nairobi Kenya” with pleasure. The paper is well written, quite simple but the results and the topic are quite interesting and actual, since they are referring to some of the more important neglected diseases in these last three decades: HIV and mental disorders (anxiety in this specific case).

Despite that, according also with the reviews received, which could be very usefule to improve the manuscript if well answered, I propose some major revisions.

Following please you can find some brief suggestions by my side.

- The title it could be clearer. I guess it’s slightly confusing. I mean, it is not clear what is being referred to. For example, it’s not clear (from the title itself) whether the Authors studied predictors for HIV or for anxiety disorder. Furthermore, from the title and the abstract should emerge more clearly that the target population are patients living with HVI on treatment with ART.

- Lines 57-58: mental health is really a human right, as stated in the WHO definition of health, but also mental health is neglected and still need more attention and interest in the public health context. Please modify this sentence.

- Line 61: please replace the term “determine”. Maybe something like “[…] to assess/estimate the prevalence of anxiety among PLHIV receiving treatment at Mbagathi Hospital in Nairobi County. Furthermore, as a secondary aim, we evaluated predictors correlated with anxiety.” Or something like that. It should be clear that the Authors aimed at evaluating the prevalence of anxiety disorder and its predictors.

- Lines 71-72: the inclusion and exclusion criteria are quite clear. I guess that Authors should explain why patients receiving mental healthcare within or outside Mbagathi Hospital were excluded from the study.

- I guess the Authors should give a better description of the catchment area from which they selected the cohort study. I believe that a better description of the Mbagathi Hospitals’ Comprehensive Care Clinic, with a description of how many patients are treated there, how many patients are assisted in general by this clinic, could give more interest and support a better comprehension.

- Line 103: “with anxiety” maybe could be replaced with “with the presence of symptoms related with moderate to severe anxiety”

- Please provide also mean and SD of descriptive statistics for quantitative variables. Or median, where more suitable.

- Line 117: “identified, 43(21%) had…”

- Lines 116-119: please use also the cutoffs of BAI in this sentence

- In table 3 insert in the header the total of patients in the group with presence of anxiety (60 I guess) and without the presence of anxiety (the other 145 patients, if I’m not wrong).

- Please control and make sure for having defined all the acronyms and abbreviations along the manuscript (sSA, TB, etc.).

Reviewers' comments:

Reviewer's Responses to Questions

**Comments to the Author**

1. Does this manuscript meet PLOS Mental Health’s publication criteria? Is the manuscript technically sound, and do the data support the conclusions? The manuscript must describe methodologically and ethically rigorous research with conclusions that are appropriately drawn based on the data presented.

Reviewer #1: Yes

Reviewer #2: Partly

2. Has the statistical analysis been performed appropriately and rigorously?

Reviewer #1: Yes

Reviewer #2: No

3. Have the authors made all data underlying the findings in their manuscript fully available (please refer to the Data Availability Statement at the start of the manuscript PDF file)?

Reviewer #1: Yes

Reviewer #2: Yes

4. Is the manuscript presented in an intelligible fashion and written in standard English?

Reviewer #1: Yes

Reviewer #2: No

5. Review Comments to the Author

Reviewer #1: I’ve read with interest the work titled “Anxiety disorder and predictors among HIV patients in Nairobi Kenya”.

The paper is original and of high interest to readers. Moreover, the objective of the paper is really interesting and actual, and this manuscript could provide a useful contribution to the literature.

I propose some minor revisions.

By section:

Introduction

1. The introduction is well written and summarizes the current state of the art well. The rationale is clear and the literature used is recent. In line 55, I would add the references to give greater support to the statement.

Materials and Methods

2. The materials and methods are generally well written. Nevertheless, I would add more details to some passages in order to make this part more informative. What is the source of the data, for example? When were they collected? Specifying the sources of the data enables you to understand a whole series of biases (e.g., selection bias and information bias). In studies such as this one that evaluate prevalence, representativeness is essential. Cross-sectional studies of diseases, traits, or other problems are therefore susceptible to selection bias. In cross-sectional studies as in any other, information bias, the inaccurate measurement or recording of a disease or characteristic, is also a key issue.

3. In line 67 it is reported “This cross-sectional study was conducted between 1st February, 2023 and 31st May, 2023”. Can you explain why this date range was chosen? Is it related to the study's rationale or to data availability? It would be beneficial to add more details to the study.

4. According to line 69, "Eligible participants had a documented HIV diagnosis"; what do you mean by "documented HIV diagnosis”? Whenever possible, it is best to report a pathology identification code. Also, when was it verified? This information is always good to provide, even if it may seem obvious. Regarding the cohort selection, I would like to add more details (also supported by the supplementary materials).

5. What is the purpose of including patients aged >18 years and excluding subjects aged 18 years?

6. The exclusion criteria are reported in lines 71-72. Also in this case, did you limit yourself to excluding the subjects who received assistance in the period 1st February, 2023 and 31st May, 2023? Or did you also exclude those who had also received assistance in previous periods (for example in the previous 5 years)? Despite the study's design, it is advisable to include these details anyway to avoid any misunderstandings on the part of the reader.

7. What do you mean by “those receiving mental healthcare”? Please specify all the services considered, the reasons why they were excluded, the period and any service identification codes (also, in this case, I would suggest adding specifications in the supplementary materials).

8. Line 72. What do you mean by “those unable to provide consent”? Adding more details would be appropriate in this case. Having a more accurate definition of these subjects allows us to understand the potential nonresponse bias. That is, any systematic difference between respondents (people who complete a survey) and non-respondents (people who do not complete a survey).To make the study generalizable and above all replicable, you should add more details in this paragraph.

9. Lines 89-97: add more details regarding the procedure through which the data were collected.

10. Lines 99-100. In this paragraph, it would be useful to indicate which descriptive statistics were used. It would also be useful to add the standard deviation to those you entered.

11. Have you used some specific procedure for the selection of the variables that were used for the estimation of the multivariate logistic regression model (i.e., backward elimination with the Akaike information criterion or modern shrinkage or penalization procedures, such as LASSO/least absolute shrinkage and selection operator, elastic net, and their variants.)?

Results

12. In general, I would always add the same number of decimals when reporting results (e.g., typo in line 113).

13. In order to simplify reading for the reader, the numbers (with any approximations) must be the same in the tables. It is recommended that you correct any typos in the following lines: (i) 124 (p = 0.027, the table shows p = 0.026); (ii) 130 (p = 0.0104, the table shows p < 0.001); (iii) 132 (p = 0.0125, the table shows p = 0.013).

Reviewer #2: This study aimed to investigate the prevalence and correlates of anxiety disorder among HIV patients receiving care and treatment at a comprehensive care clinic in Nairobi, Kenya. However, several issues need to be addressed:

-The authors did not provide an explanation for why mental health issues among People Living with HIV (PLHIV) in low‐ and middle‐income countries (LMICs) should receive deserving attention. The introduction section lacks detail and fails to contextualize the significance of mental health concerns in this population within the broader socioeconomic and healthcare landscape of LMICs.

-The study did not specify the inclusion or exclusion criteria for participant selection. Clear criteria are essential for ensuring the sample's representativeness and the study's generalizability.

- Depression and anxiety often coexist and can have overlapping symptoms. However, the study did not

---

## [Decision Letter · Decision Letter 1]

25 Apr 2024

PMEN-D-23-00061R1

Proportions and Predictors of anxiety disorder among HIV patients on ART in Nairobi Kenya

PLOS Mental Health

Dear Dr. Ngayo,

Thank you for submitting your manuscript to PLOS Mental Health. After careful consideration, we feel that it has merit but does not fully meet PLOS Mental Health’s publication criteria as it currently stands. Therefore, we invite you to submit a revised version of the manuscript that addresses the points raised during the review process.

We look forward to receiving your revised manuscript.

Kind regards,

Matteo Monzio Compagnoni, Ph.D.

Academic Editor

PLOS Mental Health

Journal Requirements:

1. We have noticed that you have a list of Supporting Information legends in your manuscript. However, there are missing files in the submission. Please upload them as separate files with the item type 'Supporting Information'. Or if the Supporting Information is no longer to be included as part of the submission please remove all reference to it within the text.

Additional Editor Comments (if provided):

I’ve read the revised version of the work titled “Proportions and Predictors of anxiety disorder among HIV patients on ART in Nairobi, Kenya” (manuscript ID PMEN-D-23-00061-R1), by Musa Otieno Ngayo, et al, with pleasure. It is a simple, good study giving an insight on the burden of anxiety disorder HIV patients treated with antiretroviral therapy in a district of Kenya. I am pleased that the Authors have addressed all the comments raised by the Reviewers in the first round of peer-review. The paper is clearly improved, and I believe that it can contribute to the literature on HIV and anxiety, specifically in Africa.

However, according also with the suggestions received from the Reviewers, I believe that some MINOR REVISIONS should be made, before being considered completely fit for the publication standards of this Journal. Indeed, after having addressed these (minor) comments, the paper should be considered as worthy of publication.

Here some minor comments (to be considered in addition to those of the Reviewers):

- Line 25: Insert in the abstract the cut-off considered for the BAI score for a patient to be considered with anxiety symptoms.

- Line 25: replace “to gather attributes” with “to collect information on characteristics”.

- Line 27: please delete the information related to the software; it is not relevant for the abstract.

- Line 34: replace “were more likely to have anxiety disorder” with “were factors associated with having anxiety disorder”.

- Line 35: “Among PLHIV receiving ART in a cunty of Kenya, there is a considerable …”

- Line 47: delete “(mental disorder)”, is redundant.

- Line 47: replace “is bidirectional” with something less secure, like “seems to be bidirectional”.

- Line 49: Authors refers to “progression” of HIV, and what about the incidence?

- Ine 51: please add a reference for the “29%”.

- Line 52: “prevalence of anxiety disorder among PLHIV…”.

- Line 99: “… study, according to the inclusion criteria, all patients …”.

- Line 102: delete “the researcher planned and organized” and use something like “The data collection process was planned and organized in accordance with the CCC in charge. The best time and day for questionnaire administration were identified to suite …”

- Line 104: “were trained to administer”.

- Line 107: “These research assistants gathered …”.

- Line 121: move “Data analysis was performed using R software version 4.1.2.” at the end of the Statistical Analysis section.

- Line 123: “was used for numeric variables”.

- Line 126: please insert a reference for the cutoffs used.

- “bivariate” and “multivariate”, instead of “bivariable” and “multivariable”.

- Please, check along the manuscript that is “Lemeshow” (reference #16) and not “LemAshow”.

- Line 102: why authors used a cut-off p-value of 0.2? Please explain that or use a reference for that choice, instead of using 0.05, for example.

- Line 137: “A total of 205 participants with a median age of 49 years”.

- Please, modify Table 1, in a more scientific template (like that adopted for Table 3, for example).

- Line 150: “… of the participants considered not anxious, or without the presence of anxiety symptoms”.

- In the Table 4, I suggest reporting only the results (aOR and 95% CI) only for the multivariate analysis, and not those obtained from the bivariate analysis. Reporting both it would be a bit redundant. Please, report these bivariate results in a supplementary table (exactly like Table 4, but supplementary).

- Line 168: “anxiety, in a county of Kenya”, I suggest to not be too general and categorical, and to always report the geographical context.

- Line 190: “anxiety symptoms”.

- Line 248: “The identified associated factors emphasize the multifaceted…”.

- I suggest to adopt a more frequent use of the impersonal third person in the sentence construction, all along the manuscript.

- I believe that a further proofread and language control should be made.

Reviewers' comments:

Reviewer's Responses to Questions

**Comments to the Author**

1. If the authors have adequately addressed your comments raised in a previous round of review and you feel that this manuscript is now acceptable for publication, you may indicate that here to bypass the “Comments to the Author” section, enter your conflict of interest statement in the “Confidential to Editor” section, and submit your "Accept" recommendation.

Reviewer #1: All comments have been addressed

Reviewer #3: All comments have been addressed

Reviewer #4: (No Response)

2. Does this manuscript meet PLOS Mental Health’s publication criteria? Is the manuscript technically sound, and do the data support the conclusions? The manuscript must describe methodologically and ethically rigorous research with conclusions that are appropriately drawn based on the data presented.

Reviewer #1: Yes

Reviewer #3: Yes

Reviewer #4: Yes

3. Has the statistical analysis been performed appropriately and rigorously?

Reviewer #1: Yes

Reviewer #3: Yes

Reviewer #4: Yes

4. Have the authors made all data underlying the findings in their manuscript fully available (please refer to the Data Availability Statement at the start of the manuscript PDF file)?

Reviewer #1: Yes

Reviewer #3: Yes

Reviewer #4: Yes

5. Is the manuscript presented in an intelligible fashion and written in standard English?

Reviewer #1: Yes

Reviewer #3: Yes

Reviewer #4: Yes

6. Review Comments to the Author

Reviewer #1: I have read the article now titled “Proportions and Predictors of Anxiety Disorder Among HIV Patients on ART in Nairobi Kenya”. I am pleased that the authors have addressed the issues raised in the previous round.

Currently, the manuscript is a well-written research paper whose main purpose is to estimate the prevalence among PLHIV receiving treatment at Mbagathi Hospital in Nairobi, Kenya. While the secondary aim is to evaluate the predictors correlated with anxiety in the same population.

I believe the manuscript meets the publication standards of the journal.

Reviewer #3: Proportions and Predictors of anxiety disorder among HIV patients on ART in Nairobi Kenya

General comments and some key concerns:

It is a good study giving an insight on the burden of anxiety disorder HIV patients on antiretroviral therapy (ART). However, below are my comments.

1. Title:

Line 1-2: Title need to be paraphrased as “Prevalence and Predictors of anxiety disorder among HIV patients on antiretroviral therapy (ART) in Nairobi Kenya” OR “Burden and Predictors of anxiety disorder among HIV patients on antiretroviral therapy (ART) in Nairobi Kenya”

Line 9: It should be written as:- Department of Pharmacy, Directorate of Clinical Services, Kiambu County, Nairobi, Kenya

2. Abstract

• Line 20: Replace “estimated” with “determined”

• Line 20: Sentence seems to be the aim but it doesn’t match with the title. It needs to be paraphrased

• Line 35: Seems to be a conclusion but it is missing the element of burden in it.

3. Introduction

• Line 65: prevalence of what??

4. Methodology

• Line 69: Materials and Methods should be replaced with “ Methodology”

• A number of important components of the methods are missing or mixed up and therefore not very clear

• Also, I suggest that the Methodology be sub-sectioned for ease of follow flow as below:

o Study design

o Study setting

o Study population

o Sample size

o Selection criteria – Inclusion and exclusion

o Sampling procedure

o Data collection tool

o Data collection procedure

o Data management and quality control

o Statistical data analysis

o Ethical consideration

• Line 107: Sentence has grammar problem

5. Results

• Line 138: Values should be reported as % (n) such as 59.5% (n=122), while 55.1% (n=113) since it is the proportion that is reported and not numbers. And this should be affected across the manuscript.

• Line 151: Sentence ……….”9) and the individual symptoms score (Table 2). Put table 2 in brackets and this should be the same across the manuscript where referenced in the text i.e. see also Line 156.

• Line 144: Sentence “………..of tuberculosis infection, and 35 (17.1%) had other comorbidities (Table 1)” is correct

• Line 153: Positive correlation is mentioned but at what level and what was the strength of the correlation?

• Line 385: Sentence …..” Table 1: Baseline characteristics of the study population (n =205 patients)” teerm population should be replaced with study participants

• In table 1, 2 and 3: Values should be reported as % (n)

• Tables should be presented as scientific tables

• Figure has no title

6. Discussion

• There is a lot of use of first and second person in the sentence constructions. I suggest that the author adopt the third person in the sentence construction.

• Demographic characteristics of the study participants are not discussed.

• How do the present findings relate to the previous similar findings from other studies

• Line 168-169: Actual values should accompany the terms majority, moderate and severe based on the actual findings from the study. Then also tables and/ or figures should be cited.

• Line 202: Sentence “……….among PLHIV. Consistent with this study Maier et al [9]…..,” et al should be written as “ et al. (9).

7. Conclusion

• It is missing in the document

Reviewer #4: This study describes a prevalence of anxiety among HIV patients in Kenya. correlations between anxiety and other factors are also described.

Abstract:

1. Line 31-32: were participants medically diagnosed with anxiety disorder? If this is from the BAI, address accordingly.

2. Line 33-34: this sentence can be made clearer. I am assuming participants do not have formal diagnosis of anxiety disorder

3. Line 35-37: this sentence can be made clearer

4. Did duration on ART affect participant anxiety levels/severity?

Introduction

1. 47: “mental disorders” is not needed

2. 51,54: missing citation; please review all and make sure to cite

3. 59-61: this sentence can be made clearer. What is meant by “interest in the public health context”

Methods

1. 72: ART need to be fully spelled out as this is the first time it is mentioned in the article (not abstract)

2. 80: capitalization

3. 86: What language wa

---

## [Editor Report · Decision Letter 2]

31 May 2024

Burden and Predictors of anxiety disorder among HIV patients on ART in Nairobi Kenya

PMEN-D-23-00061R2

Dear Dr. Ngayo,

We are pleased to inform you that your manuscript 'Burden and Predictors of anxiety disorder among HIV patients on ART in Nairobi Kenya' has been provisionally accepted for publication in PLOS Mental Health.

Best regards,

Matteo Monzio Compagnoni, Ph.D.

Academic Editor

PLOS Mental Health